# Mesoscale Variability and Water Mass Transport of the Caribbean Current Revealed by High-Resolution Glider Observations

Joseph C. Gradone<sup>1</sup>, W. Douglas Wilson<sup>2</sup>, Scott M. Glenn<sup>1</sup>, Leah N. Hopson<sup>1</sup>, Travis N. Miles<sup>1</sup>

<sup>1</sup>Center for Ocean Observing Leadership, Department of Marine and Coastal Science, Rutgers University, New Brunswick, NJ, 08901, USA

<sup>2</sup>Center for Marine and Environmental Studies, University of the Virgin Islands, St. Thomas, VI, 00802, USA *Correspondence to*: Joseph C. Gradone (jgradone@marine.rutgers.edu)

Abstract. The Caribbean Through-Flow (CTF) provides a key pathway linking the North Atlantic Subtropical Gyre and the upper limb of the Atlantic Meridional Overturning Circulation. Yet, its internal structure and variability remain poorly resolved. Autonomous underwater gliders offer a unique capability to address this gap by collecting high-resolution hydrographic and velocity observations in regions where sampling is sparse. Here, data from a glider that operated for >90 days along 69°W in summer 2024 were analyzed to investigate mesoscale-driven variability in the CTF. Two consecutive occupations of this ~600 km trans-Caribbean section revealed a sharp decline in zonal transport from -17.64 Sv to -9.22 Sv, coinciding with a shift in mesoscale activity. Rossby number and dynamic height anomaly calculations from the glider data showed a shift from flow largely in geostrophic balance during Transect #1 to increased mesoscale influence during Transect #2. Satellite altimetry spanning the full deployment suggested this shift was driven by a cyclonic eddy that passed through the northern half of the section between the timing of the two transects. Despite the large changes in transport between transect occupations, water mass analysis showed that the relative contributions from North and South Atlantic water masses remained nearly constant. Direct sampling of an anticyclonic eddy during a partial Transect #3 revealed strong temperature and salinity anomalies in the upper 200 m. These findings highlight how glider observations can resolve key features and processes governing variability in this critical inter-basin pathway and improve understanding of mesoscale influences on large-scale circulation.

#### 25 1 Introduction

40

45

Western boundary currents (WBCs) are critical components of the global climate system, facilitating the poleward transport of heat, salt, tracers, and momentum. In the western tropical Atlantic, the Caribbean Through-Flow (CTF), composed largely of the Caribbean Current, represents a major upper-ocean conduit linking the North Atlantic Subtropical Gyre (NASTG) and the upper limb of the Atlantic Meridional Overturning Circulation (AMOC). The CTF exports upper-layer waters through the Yucatan Straits, supplying the Florida Current and, subsequently, the Gulf Stream (Johns et al., 2002; Rhein et al., 2005; Gradone et al., 2025b). As such, the Caribbean Sea constitutes a critical chokepoint for transforming and redistributing water masses, influencing interhemispheric circulation.

Despite its importance, the structure and variability of the CTF remain insufficiently resolved, particularly in the subsurface. Mesoscale features and processes, such as eddies, jets, and countercurrents, likely play a substantial role in modulating transport and mixing but are poorly captured by traditional observing systems. The A22 section along 66°W in the Caribbean has been occupied four times from 1997 to 2021 by several different hydrographic programs, including WOCE (World Ocean Circulation Experiment), CLIVAR (Climate Variability and Predictability), and GO-SHIP (Global Ocean Ship-based Hydrographic Investigations Program). While these programs have improved understanding of large-scale circulation, the station spacing ranging from 25–50 km (Menezes, 2022) is insufficient to resolve mesoscale features, which typically exhibit scales of 10–100 km.

Autonomous underwater gliders now offer the capability to resolve oceanic processes at much finer spatial and temporal resolution, with sampling scales on the order of 2 km horizontal and 1 m vertically. These platforms have proven particularly effective in the Caribbean, where mesoscale eddies are known to introduce both transient and persistent variability in the current structure (Gradone et al., 2023). While observing systems like Argo have enabled the detection of large-scale water mass changes in the CTF (Gradone et al., 2025b), the limited spatial and temporal coverage of subsurface velocity measurements makes it difficult to assess corresponding changes in volumetric transport. Near 69°W, observations suggest the Caribbean Current ranges from -13.7 Sv to -26.3 Sv (1 Sverdrup = 10<sup>6</sup> m³ s⁻¹) (Johns et al., 2002; Casanova-Masjoan et al., 2018), a discrepancy as large as the lower transport estimate itself. To close the budget with the estimated ~30 Sv mean transport exiting the Yucatan Straits (Rousset and Beal, 2014), this flow must be augmented by ~6–9 Sv from the Windward Passage (Figure 1) (Wunsch and Grant, 1982; Nelepo et al., 1978; Roemmich, 1981; Johns et al., 2002). The Yucatan Straits, therefore, provide a critical benchmark: they not only constrain the mass balance of the CTF but also supply 88–100 % of the source waters of the Florida Current (Rousset and Beal, 2014), linking Caribbean variability directly to the circulation of the greater Atlantic. Resolving the wide uncertainty in Caribbean Current transport (Johns et al., 2002; Casanova-Masjoan et al., 2018) is therefore essential for understanding the dynamics and variability of the CTF and its role in Atlantic circulation.

In this study, high-resolution observations from an autonomous underwater glider were used to investigate the internal dynamics of the Caribbean Through-Flow near 69°W. Unlike traditional ship-based sampling along sections like A22, which lack the resolution to capture mesoscale features, this dataset provided detailed hydrography and subsurface velocity measurements at scales capable of resolving mesoscale-driven variability. First, the variability in zonal transport between repeat glider transects was quantified, and its sensitivity to mesoscale structure was assessed. Second, the relative contributions of North and South Atlantic water masses to the Caribbean Current transport were examined, and the consistency of these distributions under contrasting dynamical conditions was discussed. Finally, the influence of mesoscale eddies on current structure and water mass properties was explored by combining in situ measurements with satellite altimetry and dynamical diagnostics such as Rossby number and temperature and salinity anomalies. These findings provide new insight into the internal dynamics of the Caribbean Current and underscore the necessity of resolving eddy-driven processes in both observational and modeling frameworks.

**Figure 1:** Map of the Caribbean region with glider track in black, arrows indicating transect direction, and WOCE A22 stations as grey circles. Transect #1 (white) and Transect #2 (yellow) are offset from the half Transect #3 (orange) along -69°W for visualization.

#### 2 Data and Methods

# 2.1 Autonomous underwater glider data

The observations used in this study were collected using a Teledyne Webb Research Slocum glider (Schofield et al., 2007). A deep (1,000 m rated) second-generation Slocum glider (RU29) sampled an approximately 600 km transect along 69°W from the Dominican Republic to Curação, traveling over 2,300 km (2.5 transects total, with each transect taking ~3 weeks) over 95 days (Figure 1). This glider was equipped with a Sea-Bird Scientific pumped conductivity, temperature, and depth (CTD) sensor and a 1-MHz Nortek AD2CP. The AD2CP is a four-beam acoustic current profiler configured to sample with 15 cells

100

of 2-meter vertical resolution, averaging four pings for 1 second every 5 seconds. The CTD data were sampled every 2 seconds throughout complete dives and climbs. Potential density ( $\sigma_{\theta}$ , from here on: density) was calculated from conservative temperature ( $\Theta$ , from here on: temperature) and absolute salinity ( $S_A$ , from here on: salinity) measurements using the TEOS-10 standard from the Gibbs Sea Water (GSW) Python package (Roquet et al., 2015; Mcdougall et al., 2021). Individual dives and climbs were treated as separate profiles, and all hydrographic data were averaged into 2-meter vertical depth bins. Dynamic height anomaly (DHA) was computed by vertically integrating the specific volume anomaly relative to 990 decibars by:

$$DHA(p) = g^{-1} \int_{990}^{p} \delta v(p) dp$$

where  $\delta v$  is the specific volume anomaly, g is the gravitational acceleration, and p is the pressure derived from depth. The reference of 990 decibars was chosen for consistency with the majority of glider profiles reaching at least this depth.

Processing velocity measurements from glider-mounted current profilers involved a series of quality control and correction

## 5 2.1.1 Transport from glider-mounted acoustic doppler profiler derived horizontal water velocity

steps to ensure accurate estimates of ocean currents. The processing steps include quality control, mapping beam velocities to vertical bins relative to the glider, correcting for glider orientation to assign level true depths, performing a coordinate transformation from beam coordinates to East-North-Up (ENU), and deriving absolute horizontal water velocities. The reader is referred to the methodology in Gradone et al. (2023) for the specifics of the processing steps performed in this analysis. Absolute horizontal water velocities were interpolated onto a regular latitude–depth grid to enable spatially consistent comparison of velocity data along the two transects between the Dominican Republic and Curação. A constant longitude of –69.0° was used for all grid points because glider movement was primarily north–south and east–west drift was minimal. The latitude grid was constructed with a uniform spacing of 5.5 km, chosen to include approximately 1–2 glider profiles per bin based on the gliders' typical horizontal speed (~1 km hr<sup>-1</sup>) and segment length (~3 km). At each depth level, E–W and N–S velocity observations were interpolated to this grid using 1D linear interpolation (Figure 2). This gridding approach spatially smooths the velocity field while preserving large-scale structure. After gridding, transport for each glider segment (latitude bin) was calculated by multiplying the E-W velocity at each depth by the vertical bin thickness and the latitude bin length, then integrating over depth to obtain transport per segment in Sverdrups. These segment transports were subsequently summed along the entire transect to yield the total transport.

Finally, Rossby number  $(R_0)$  was estimated from the velocity profile data for each transect as:

$$R_O = \frac{\zeta}{f} = \frac{\frac{\delta v}{\delta x} - \frac{\delta u}{\delta y}}{f}$$

where  $\zeta$  represents relative vorticity and f represents the local Coriolis parameter. The Rossby number calculation was simplified by eliminating the  $\frac{\delta v}{\delta x}$  term as the glider transects sampled only in the meridional direction.

110

# 105 2.1.2 Water mass analysis

Water mass analysis used temperature and salinity data to distinguish between South Atlantic Water (SAW) and North Atlantic Water (NAW) fractions within the CTF. Followed the approach of Rhein et al. (2005), this method uses an isopycnal mixing approach with least-squares fitting to estimate the relative contributions of the two source waters to the observed temperature and salinity at each density level, based on representative source water profiles (Figure 3). Representative source water properties were taken from World Ocean Atlas 2018 (Locarnini, 2019; Zweng, 2019) and converted to conservative temperature and absolute salinity using TEOS-10 standards. The isopycnal mixing method followed:

$$x_{SA}T_{SA} + x_{NA}T_{NA} - T_{obs} = R_T$$

$$x_{SA}S_{SA} + x_{NA}S_{NA} - S_{obs} = R_S$$

$$x_{SA} + x_{NA} - 1 = R_{MC}$$

where (T<sub>SA</sub>, S<sub>SA</sub>) and (T<sub>NA</sub>, S<sub>NA</sub>) represent the temperature and salinity definitions for the representative SAW and NAW types; (T<sub>obs</sub>, S<sub>obs</sub>) are the observations; (x<sub>SA</sub>, x<sub>NA</sub>) are the fractional relative contributions from each source; and (R<sub>T</sub>, R<sub>S</sub>, R<sub>MC</sub>) are the residuals minimized through least squares fitting. The fractional source contributions were then multiplied by the corresponding measured velocity and vertically and horizontally integrated to estimate transport by water mass. This approach parallels optimum multiparameter methods but is simplified to two water mass endmembers for a determined system of equations. For a comprehensive description of the methodology, including assumptions, source water definitions, and detailed equations, the reader is referred to Gradone et al. (2023). The Python package PYOMPA (version 0.3) is adapted for this analysis (Shrikumar, 2021). The surface layer ( $\sigma_{\theta}$  

Although the glider did not complete a full third transect, it conducted detailed in situ sampling of an anticyclonic eddy during its partial transit. During this period, the glider distance to the eddy center was calculated from the glider's location and the corresponding eddy location from the *py-eddy-tracker* output on that day. The differential anomaly method (Simpson et al., 1984) was used to quantify eddy-induced water mass anomalies. This method isolates temperature and salinity deviations from a reference profile caused by isopycnal displacements at specific depths by:

$$T_a(r,z) = T_e(r,z) - T_r(z)$$

$$S_a(r,z) = S_e(r,z) - S_r(z)$$

where eddy temperature (salinity) anomaly  $T_a(r,z)$  at a radial distance r from the eddy center and depth z is defined as the difference between the observed temperature (salinity)  $T_e(r,z)$  and a reference temperature (salinity) profile  $T_r(z)$ . The reference profiles were constructed by averaging all glider data outside the eddy diameter but within a 50 km buffer zone beyond its outer boundary. A total of 117 profiles met this criterion.

**Table 1:** Mean anticyclonic eddy characteristics derived from satellite altimetry and glider data. Numbers in parentheses represent values for the cyclonic eddy not directly sampled by the glider between the two transects.

| Characteristic          | Symbol                                                          | Magnitude                                                  |
|-------------------------|-----------------------------------------------------------------|------------------------------------------------------------|
| Sea level anomaly       | A                                                               | 0.3 m (-0.1 m)                                             |
| Swirl velocity          | U                                                               | 0.12 m s <sup>-1</sup>                                     |
| Translation velocity    | V                                                               | 0.32-0.48 m s <sup>-1</sup> (0.24-0.63 m s <sup>-1</sup> ) |
| Depth scale             | D                                                               | 100-200 m                                                  |
| Radius                  | R                                                               | 160 km (100 km)                                            |
| Rossby number           | $Ro = \frac{U}{f_o R}$                                          | 0.005                                                      |
| Brunt-Väisälä frequency | $N = \sqrt{-\frac{g}{\rho_o} \frac{\partial \rho}{\partial z}}$ | 0.005-0.02 s <sup>-1</sup>                                 |
| Burger number           | $Bu = \frac{N^2 D^2}{f_o^2 R^2}$                                | 0.01                                                       |

## 3. Results

#### 3.1 Current Structure and Transport

Figure 2 shows E-W and N-S velocity derived from the glider-mounted AD2CP for the two transects from the Dominican Republic to the north and Curação to the south. Though the flow was generally westward during both transects, the Caribbean Current exhibited significant variability in vertical structure and magnitude along this longitude. A strong westward jet existed around 13°N during both transects, with surface and subsurface speeds exceeding -0.3 m s<sup>-1</sup> during Transect #1. Between 16°N and 17°N, strong westward flow existed during Transect #1 but was absent and partially reversed during Transect #2. A weak but persistent countercurrent was observed in the center of each transect, around ~14°N-15°N. The N-S velocity exhibited less

variability within a given transect than the E-W velocity, but was distinctly more variable between transects. Transect #1 showed surface-intensified southward flow near 15°N, 16°N, and 17.5°N. Glider direction of travel was southward during Transect #1 and northward during Transect #2. As the glider approached the southern end of Transect #1 near ~13°N, there was a stronger northward component to the N-S velocity. Then, during Transect #2, the flow had a consistent northward component until the glider was north of 17°N. Total zonal transport across this section was estimated as -17.64 Sv for Transect #1 and -9.22 Sv for Transect #2. The potential driver of the variability in both current structure and total transport is examined in the Discussion section.

Figure 2: E-W (top row) and N-S (bottom row) velocity derived from glider-mounted AD2CP from Transect #1 (A & C) and Transect #2 (B & D).

# 3.2 Water Mass Analysis and Transport



The highly stratified water mass structure found in the Caribbean was evident in the temperature and salinity profiles shown in Figure 3. Marked by warm temperatures and comparatively lower salinity, Caribbean Surface Water (CSW) dominated the surface layer (Morrison and Nowlin Jr., 1982). Just below the CSW, Subtropical Underwater (STUW) was identified as a subsurface salinity maximum in the depth range of 90-160 meters (Wüst, 1963). The subsurface salinity maximum became shallower and saltier moving from north to south. Subtropical Mode Waters (STMW), also known as Sargasso Sea Water and 18°C Water (Worthington, 1958), were found below STUW with temperature and salinity properties generally warmer and





saltier in the northern Caribbean. Central Waters (CW) overlapped with the STMW and extended over a significant depth range in the Caribbean, and were easily identifiable by their linear T/S relationship (Montgomery, 1938; Iselin, 1939). As the maximum depth sampled by the glider was ~990 meters, the deepest water mass observed was Antarctic Intermediate Water, first identified in the Caribbean by Wüst (1963).

Transport of NAW and SAW per water mass class was calculated following the isopycnal water mass analysis (Figure 4). Including the surface layer, the total SAW (NAW) transport was -8.35 Sv (-9.29 Sv) and -4.59 Sv (-4.63 Sv) for the first and second transect, respectively. The surface layer accounted for -2.71 Sv and 1.24 Sv of SAW transport, which represents 15% and 13% of the total transport, for the first and second transect. Note, the surface layer transport during Transect #2 is positive, indicating net transport eastward. Despite the decrease in total transport by a factor of ~2 from the first transect to the second, there was comparatively minimal variation in the ratio of SAW:NAW transport. Transect #1 transport was comprised of 53% NAW and 47% SAW, whereas Transect #2 shifted to 50% NAW and 50% SAW. Excluding the surface layer, the percentage of SAW transport generally increased moving from STMWs down to IWs. For clarity, only the percentage of SAW is reported here, as the percentage of NAW can be inferred as the remainder. Though other water masses may contribute to the transport in these layers, an assumption and limitation of the methodology is that the observed properties were derived from linear mixing of two water masses. The STMWs were overwhelmingly dominated by NAW, with SAW constituting ~8-13% of the transport in this layer. The 13% SAW derived for Transect #2 represents a net eastward transport as well. Interestingly, the percentage of SAW transport relative to the total transport in the uCW layer was ~21% during Transect #1, but increased to 50%, albeit with a net transport nearly equal to zero, during Transect #2. For both transects, the percentage of SAW transport relative to the total transport in the lCW and IW layers was nearly constant at ~45%, and ~53%.

**Figure 3:** Conservative Temperature (A) and Absolute Salinity (B) profiles from the two transects colored by latitude. Temperature-Salinity diagram (C) of same data with representative source water mass profiles from World Ocean Atlas 2018 following Rhein et al. (2005) for the South Atlantic (blue) and North Atlantic (orange). Relevant water masses labeled where SW: Surface Waters above 1024.5 kg m<sup>-3</sup>, STUW: Subtropical Under Water core at 1025.6 kg m<sup>-3</sup>, STMW: Subtropical Mode Water core at 1026.5 kg m<sup>-3</sup>, and IW: Intermediate Water core at 1027.3 kg m<sup>-3</sup>.

Figure 4: Transport of South Atlantic Water (blue) and North Atlantic Water (orange) in the major water masses from the isopycnal water mass analysis for the two transects. The surface waters are hatched in blue as it was not included in the water mass analysis.

## 3.3 Anticyclonic Eddy Sampling





The glider sampled to within approximately 15 km of an anticyclonic eddy during the third (half) transect, collecting 793 profiles within the eddy diameter over 60 days. Figure 5 shows a detailed sampling of this eddy, highlighting temperature and salinity anomalies alongside E-W and N-S AD2CP velocities and the glider's position relative to a sea-level anomaly estimated from AVISO. Temperature and salinity anomalies increase considerably in the eddy interior relative to a reference profile constructed from averaging 117 glider profiles outside the eddy but within a 50 km buffer zone beyond its outer boundary. Within the inner 45 km, the maximum conservative temperature anomaly was 5.91°C at 153 meters. On average, the conservative temperature anomaly was 3.22 ± 1.46°C in the upper 200 meters within the inner 45 km. The absolute salinity anomaly in the eddy center exhibited a different structure, largely due to the presence of the subsurface salinity maximum. Within the inner 45 km, the maximum negative absolute salinity anomaly was -0.92 g kg<sup>-1</sup> at 101 meters, while the maximum positive absolute salinity anomaly was 0.58 g kg<sup>-1</sup> at 195 meters. Figure 6 shows how the glider likely missed sampling the direct center of the anticyclonic eddy but still managed to collect a few profiles on the southern side of the center. The velocity fields support this, showing strong eastward (clockwise) flow as the glider samples the northern side of the eddy, followed by a brief period of westward flow beginning around 20 June 2024, when the glider's distance from the eddy center started to increase again. The brief period of westward flow also corresponds with an increase in a southward component to the current, further supporting the glider's likely sampling of the southeastern quadrant of the eddy.

Table 1 details the eddy characteristics of the anticyclonic eddy directly sampled by the glider. The low Rossby and Burger numbers suggest the anticyclonic eddy was largely in geostrophic balance. Estimates of the sea level anomaly, radius, and

translation speed of a cyclonic eddy that passed through the region between the two transects (Figure 6), and therefore not directly sampled by the glider, are also included in Table 1 for use in the Discussion section. The estimated translation speed for each eddy was comparable to the mean zonal velocity of the Caribbean Current (Richardson, 2005).

**Figure 5:** (a) Sea level anomaly above glider RU29 and (b) distance from RU29 to eddy center determined by *py-eddy-tracker* algorithm and RU29 heading, where 0° represents travel to the north. (c) Eddy induced Conservative Temperature anomaly and (d) Absolute Salinity

anomaly relative to nearby reference profiles outside of eddy influence. (e) E-W and (f) N-S velocity derived from glider-mounted AD2CP.

**Figure 6:** Time-series of glider track (grey) against latitude overlying daily mean AVISO sea level anomaly. Relative magnitude of E-W transport per glider segment as colored vectors for the full track (grey), Transect #1 (blue), and Transect #2 (orange). Arrows to the left indicate westward transport and arrows to the right indicate eastward transport.

#### 4. Discussion



## 4.1 Transport Variability

These glider-based observations provide a new perspective on long-standing uncertainty in Caribbean Current transport near 69°W. Zonal transport estimates ranged from -17.64 Sv on Transect #1 to -9.22 Sv on Transect #2, values that fall within but also extend the range of previous ship-based measurements (-13.7 to -26.3 Sv) (Johns et al., 2002; Casanova-Masjoan et al., 2018). This overlap underscores agreement in variability while highlighting the persistent difficulty in constraining the mean strength of the current. This variability is examined in detail in the following section, and potential drivers of the discrepancy in mean values are considered. Given the role of the Caribbean Current in closing the mass budget into the Yucatan Straits and ultimately supplying the Florida Current, resolving these uncertainties is critical for linking Caribbean dynamics to the larger Atlantic circulation.

The vertical structure of the mean  $R_0$  derived from ADCP measurements was examined (Figure 7) to investigate the mechanisms behind the transport discrepancy between transects. During Transect #1, mean  $R_0$  values remained below or very

near 0.1 through the entire water column, consistent with large-scale geostrophic balance. In contrast, Transect #2 showed mean  $R_0 > 0.1$  for nearly the entire profile and standard deviations in the 0-200 m and 400-900 m ranges approaching 1, indicating substantial variability and dynamically significant ageostrophic motions. Because ageostrophic eddies can redistribute momentum and weaken the large-scale current, this suggests that enhanced mesoscale variability during Transect #2 was the primary driver of the reduced transport (Figures 7 & 8).

Figure 7: Mean Rossby number profiles for Transects #1 and #2 shaded by +/- 1 Standard deviations

Additional evidence comes from comparing glider-derived DHA with integrated transport per glider segment (Figure 8 ). The consistent agreement between glider-derived DHA and AVISO ADT (Supplemental Figure 1) is expected, as the former captures the baroclinic structure and the latter captures both baroclinic and barotropic contributions to sea level height. However, there are locations where depth-integrated transport per glider segment does not relate to the glider-DHA, particularly the northern half of Transect #1, where strong barotropic flow (Figure 8) coincides with  $R_0 

tighter correspondence between depth-integrated transport per glider segment and glider-derived DHA in Transect #2 over this same region reflects reduced barotropic influence and a stronger imprint of mesoscale variability on the flow field.

**Figure 8:** Transport per glider segment (dive and climb pair) across transect latitude for Transect #1 (solid blue line) and #2 (dashed blue line). Dynamic height anomaly derived from glider hydrographic measurements for each glider segment for Transect #1 (solid orange line) and #2 (dashed orange line).

A third factor arises from observational resolution. Historical ship-based sections in the Caribbean have typically sampled at horizontal spacings of 25–50 km, too coarse to resolve the mesoscale features revealed here to strongly influence transport. In contrast, the gliders provide effective resolution of ~5 km, capturing much finer-scale variability. When the glider subsurface velocity observations are interpolated to the coarser station spacing of the A22 section, the resulting transport estimates are biased by nearly a factor of two, even reversing the relative magnitudes of transport for Transects #1 and #2. This sensitivity underscores how under sampling can alias mesoscale variability, producing the wide spread of mean transport estimates reported over the past three decades.

## 4.2 Water Mass Transport


Following the isopycnal water mass analysis of Rhein et al. (2005), glider observations revealed a SAW transport of approximately -8.35 Sv and -4.59 Sv for the first and second transects. Interestingly, there was minimal variation in the ratio of SAW:NAW, changing only 3% despite the factor of 2 change in bulk transport. If these SAW transport estimates are taken as accurate, an additional ~8.65-12.41 Sv of SAW must flow through Windward Passage to close the ~17 Sv transport budget corresponding to the estimated strength of the AMOC (Frajka-Williams et al., 2019). This is unlikely as most of the SAW inflow is thought to be concentrated in the southeastern Caribbean passages (Kirchner et al., 2009; Rhein et al., 2005), and the northern passages exhibit stronger NAW inflow (Kirchner et al., 2009; Gradone et al., 2023). Notably, however, the 47-50% SAW fraction observed along 69°W closely matches the 45% Schmitz and Richardson (1991) reported for the Florida Current.







despite their estimate being based on only 13 Sv of SAW. Even considering the highest mean transport estimate from ship-based observations near 69°W of -26.3 Sv (Casanova-Masjoan et al., 2018) and 50% SAW would only account for -13.15 Sv of SAW. A likely alternative explanation for the observed discrepancies is a reassessment of the water mass analysis methodology, which cannot be applied in the surface layer (Schmitz and Richardson, 1991; Schott et al., 1998; Hellweger, 2002; Rhein et al., 2005) and neglects contributions from high-salinity South Atlantic waters (Zhang et al., 2003; Rhein et al., 2005; Kirchner et al., 2008).

Additionally, it is important to acknowledge that the glider-based transport estimates, particularly for Transect #2, where mesoscale variability was pronounced, may underestimate the mean flow. This underestimation likely arises from navigational constraints that prevent sampling the full width of the Caribbean Current and from the gliders' depth limitation to ~1000 m, which does not capture the entire vertical structure of the flow. Consequently, while the observed SAW:NAW ratio is informative, the absolute transport values should be interpreted as a lower bound on the total SAW contribution to the AMOC and viewed as motivation for improved methodological approaches to better constrain water mass transport in this region.

## 4.3 Eddy Influence on Water Mass Structure

Mesoscale eddies are key components of the climate system, redistributing tracers and momentum and influencing ocean circulation, heat and nutrient transport, carbon sequestration, and gas exchange (Beech et al., 2022). The CTF eddy field has been well characterized in terms of formation (Carton and Chao, 1999; Jouanno et al., 2009), propagation (Richardson, 2005), dissipation (Carton and Chao, 1999), variability (López-Álzate et al., 2022), and climatic forcing (Huang et al., 2023), and is thought to intensify from east to west due to the westward growth of baroclinic instabilities (Carton and Chao, 1999; Jouanno et al., 2008). However, how these eddies modify subsurface water mass structure in the CTF remains unresolved (Gradone et al., 2025a).

The detailed sampling of an anticyclonic eddy during the third transect quantifies how mesoscale structures modify local water mass properties and structure. The eddy generated strong temperature and salinity anomalies in the upper 200 m, along with pronounced clockwise velocity components consistent with an anticyclonic circulation. In contrast, the SLA field suggests that a cyclonic eddy passed through the region between the first and second transects (Table 1), likely contributing to the higher Rossby numbers observed during Transect #2, which increased mesoscale variability. Cyclonic eddies, being more baroclinic and nonlinear than their anticyclonic counterparts, can oppose the mean flow, enhance vertical shear, and redistribute energy from the mean current (Perret et al., 2011). By contrast, anticyclonic eddies, often more barotropic, can reinforce the mean flow locally or broaden the current without substantially decreasing net transport. These observations underscore the significant role of mesoscale eddies, both cyclonic and anticyclonic, in modulating the Caribbean Current, influencing local water mass properties, and generating variability in transport estimates.

## 5. Conclusion





High-resolution glider observations from repeat transects along ~69°W provide new insight into mesoscale-driven variability in the Caribbean Through-Flow. Zonal transport declined sharply from -17.64 Sv on the first transect to -9.22 Sv on the second, coinciding with a shift from flow largely in geostrophic balance to conditions dominated by mesoscale variability. Rossby number calculations from glider hydrographic and subsurface velocity data support this increased variability, and satellite altimetry data spanning the full deployment suggest a cyclonic eddy passing through the northern half of the section between transects was a key driver. Sampling of an anticyclonic eddy during a partial third transect further illustrates the impact of mesoscale features, producing strong thermal and salinity anomalies in the upper 200 m. Despite these circulation changes, there was minimal variation in the relative contributions of North and South Atlantic water masses to the total transport. These results underscore how high-resolution glider observations, combined with altimetry, can resolve the processes controlling transport variability and water mass structure in this critical pathway linking the NASTG and the upper limb of the AMOC. However, further advancements in observational methodology are necessary to fully and accurately constrain the pathways of water mass circulation within the upper limb of the AMOC. One approach to achieving this requires measurements of at least one nutrient that covaries stoichiometrically with dissolved, tailored to the number of distinct source waters, which can then be used in an extended optimum multiparameter (eOMP) analysis (Poole and Tomczak, 1999; Tomczak and Large, 1989; Valencia-Gasti et al., 2022). Other observational approaches for determining water mass origins remain limited, making it challenging to resolve circulation pathways confidently. Current observing systems and sensor technologies are insufficient for routinely acquiring the necessary data, highlighting the urgent need to develop new, more accessible nutrient sensors. Addressing these gaps will be essential for enabling high-resolution, widespread monitoring and improving understanding of the dynamics of large-scale ocean circulation.

## **Code Availability**

The code for this analysis can be found in this GitHub repository https://github.com/JGradone/RU29 69W.

## **Data Availability**

The data for this analysis can be found in this Zenodo repository <a href="https://doi.org/10.5281/zenodo.17352033">https://doi.org/10.5281/zenodo.17352033</a>.

#### 330 Author Contributions

J.C.G, W.D.W., and T.N.M. conceptualized the study. J.C.G. performed the data curation and formal analysis and wrote the manuscript draft. J.C.G., W.D.W., T.N.M., L.N.H. and S. reviewed and edited the manuscript

# Acknowledgements

J.C.G., W.D.W., and T.N.M. acknowledge the National Science Foundation Division of Ocean Sciences (NSF OCE-2421622)
for support during this paper's analysis and writing. W.D.W acknowledges Support from VI-EPSCoR NSF award OIA#
1946412. All authors acknowledge the G. Unger Vetlesen Foundation for the many years of support in conducting exploratory and capacity-building research across the greater Caribbean region.

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
