# Peer review of "Mesoscale Variability and Water Mass Transport of the Caribbean Current Revealed by High-Resolution Glider Observations"

_EGUsphere, 2025_

## Referee Comment (RC2)

Review of « *Mesocale Variability and Water Mass Transport of the Caribbean Current Reveald by High-Resolution Glider Observations* » by Joseph C. Gradone et al.

By Mathieu Gentil ([mathieu.gentil@utoulouse.fr](mathieu.gentil@utoulouse.fr))

This paper presents a >90-day glider mission along 69°W aimed at characterizing the mesoscale variability of the Caribbean Through-Flow and its impact on zonal transport and water-mass structure. Using high-resolution hydrographic and velocity data from successive glider transects, complemented by satellite altimetry, the authors report a strong reduction in zonal transport between glider transects and attribute this change to a shift in mesoscale activity. A water-mass analysis indicates that the relative fractions of South Atlantic and North Atlantic waters remain nearly unchanged between transects despite the large difference in transport, and the authors attribute this apparent stability to mesoscale control of the flow, further illustrated by the strong temperature and salinity anomalies observed within the anticyclonic eddy sampled during the third transect. The study intends to highlight the value of combining glider observations with altimetry to better resolve transport variability in this key inter-basin pathway.

This paper relies on an exceptional and still rare dataset: more than 90 days of glider–ADCP measurements providing hydrographic and velocity observations at ~5 km resolution in a key region for the North Atlantic water-mass pathways. Such a dataset is inherently valuable and deserves publication, as it offers a unique view of the hydrological and dynamical structure of the Caribbean Through-Flow at scales approaching the submesoscale. However, in its current form, the manuscript remains largely descriptive and several aspects of the dynamical interpretation require clarification and deeper analysis. Given the scientific importance of the topic and the quality of the observations, I believe the study should be published, but only after major revisions that strengthen the physical interpretation and improve the overall clarity of the work. I have chosen to identify myself in case the authors wish to contact me regarding any of the issues raised below.
* * *
**C1**— The manuscript attributes the strong reduction in zonal transport between Transect #1 and #2 to enhanced mesoscale activity, largely based on the Rossby number profiles shown in Figure 7. However, the glider sampling geometry does not allow estimation of $\partial v/\partial x$, so the computed Ro represents only a partial measure of shear and should be interpreted as a qualitative intensity proxy rather than an indicator of ageostrophic imbalance. In its current form, the Ro(z) structure is therefore not sufficient to support the proposed dynamical interpretation.

A more robust way to assess the role of mesoscale variability would be to quantify the mesoscale component of the velocity field directly. Computing the local Rossby radius of deformation at these latitudes would provide an objective mesoscale scale. Applying a spatial filter to separate large-scale and mesoscale currents would allow the authors to derive a profile of $u'(z)$ for each transect and, consequently, a mesoscale transport component $T'$. Comparing the vertical structure and magnitude of $u'$ between Transects #1 and #2 would provide a clearer and more quantitative assessment of mesoscale influence than Ro alone. Clarifying these aspects would significantly strengthen the physical argument linking mesoscale variability to the observed transport change.

**C2--** The interpretation of the water-mass transports contains a conceptual point that would benefit from clarification. The manuscript notes that discrepancies in SAW transports may stem from limitations of the method in the surface layer (Section 4.2), yet the analysis itself assigns this entire layer ($\sigma_3 < 24.5$) to 100% SAW. It is therefore unclear how surface-layer uncertainties could explain the reported mismatch. It would also be useful to indicate explicitly in the method whether the analysis is consistently applied below the mixed layer, where the method is formally valid.

Beyond the SAW:NAW fractions—which are indeed relevant for understanding the large-scale pathways feeding the AMOC—Figure 4 reveals a second, important aspect: the transport associated with the individual water-mass classes (SMW, uCW, ICW, IW) changes substantially between Transect #1 and #2, yet this variability is not discussed. Such differences could reflect mesoscale-driven deformation of the water-mass structure, for instance through changes in layer thickness, vertical displacement, isopycnal tilting, or enhanced lateral and vertical mixing, mechanisms documented for similar structures in our last paper (https://agupubs.onlinelibrary.wiley.com/doi/full/10.1029/2024GL110845). However, this remains to be demonstrated with the present dataset. Exploring how the thickness or vertical extent of each water-mass layer evolves—particularly along Transect #3 as the glider approaches the anticyclonic eddy—could provide valuable insight into mesoscale water-mass transformation processes and would considerably strengthen the dynamical interpretation of the observed transport changes.

**C3--** Section 4.3 discusses how mesoscale eddies modify water-mass structure, yet the current analysis remains essentially descriptive and does not demonstrate how the sampled eddies affect the vertical structure or layer distribution of the water masses. In particular, the link between the eddies inferred from altimetry and the observed transport variability would benefit from a more precise dynamical context.

Figure 6 shows a sea-level anomaly field averaged over the full 90-day deployment, which smooths out most mesoscale variability and makes it difficult to relate individual transects to specific eddy conditions. For evaluating how mesoscale structures influenced each occupation of the section, it would be more informative to show altimetric fields averaged over a time window appropriate to each transect (e.g., 15–30 days). Presenting three such maps—one for each transect—together with the glider track and surface geostrophic currents would provide a

clearer picture of the mesoscale environment sampled during each occupation. This would allow the authors to directly assess whether the reduced transport in Transect #2 corresponds to the presence of a cyclonic eddy, and whether the anomalies observed in Transect #3 reflect the glider's approach to the anticyclonic eddy. Providing this dynamical context would significantly strengthen the discussion of how eddies modulate transport and water-mass structure.

**C4--** Because much of the manuscript's interpretation relies on the glider's ability to quantify absolute velocity and transport, it would be helpful for the authors to include a brief discussion of the uncertainties associated with the ADCP-derived currents. As the glider dives towards 1000 m, the signal-to-noise ratio naturally decreases, and the reduction in percent-good and correlation values can introduce non-negligible uncertainty in the deep velocity estimates. Since the transport calculations integrate velocity over the full water column, even moderate biases or noise at depth can influence the resulting section-integrated transport, particularly when comparing differences between transects.

Providing an estimate of the typical uncertainty in the ADCP velocities—based on instrument specifications, percent-good profiles, or standard deviations of repeated measurements—and discussing how this uncertainty compares with the environmental variability of the transport would considerably strengthen the robustness of the transport interpretation. Even a brief quantitative or qualitative assessment would help contextualize the reported transport differences and reinforce the scientific results of the paper.

**C5--** Since the manuscript aims to link hydrodynamic variability with water-mass structure, it would be helpful for the figures to reflect this connection more clearly. In particular, adding the key isopycnals that define the water-mass layers (Figure 3) onto the velocity sections in Figures 2 and 5 would greatly improve readability. This simple addition would allow the reader to directly relate current structure to the underlying density field and better follow the physical interpretation.

Below are my minor comments :

Figure 1 : Please indicate the source of the bathymetry used in the map (e.g., GEBCO, ETOPO1, etc). It would also be helpful to include the year (or period) of the WOCE stations shown on the figure, so that the reader can easily compare their temporal context with that of the glider transects.

l.101 : The manuscript introduces the Rossby number but does not explain what information this diagnostic is intended to provide in the context of the study. Adding one sentence clarifying its physical meaning (e.g., measure of geostrophic balance vs. ageostrophic influence) would help non-expert readers understand its relevance.

l.145 : Table 1 reports several dynamical quantities (e.g., Brunt–Väisälä frequency, Burger number) for the anticyclonic eddy, but these diagnostics are not discussed anywhere in the manuscript. If included, their physical relevance and interpretation should be addressed in the text; otherwise, they should be removed.

Figure 2 : The colorbar in Figure 2 is inverted relative to standard oceanographic conventions (i.e., positive values in red and negative in blue), which is the convention used in Figure 5e–f. Adopting a consistent color convention across figures would greatly improve readability. In addition, adding isopycnals (and the MLD) along the section would help the reader relate the velocity structure to the underlying water masses. Finally, including an arrow at the top of the figure indicating the glider's direction of travel would clarify the continuity of the section.

Section 3.2 introduces several water masses (CSW, CW, STMW, STUW, uCW, lCW), but some of these (e.g., uCW, ICW/lCW) are not defined, and the notation appears inconsistent (e.g., "lCW" at line 190 vs. "ICW" in Figure 4). In addition, the water masses shown in the T–S diagram (Figure 3) do not match those presented in the transport decomposition (Figure 4). For clarity, it would be important to (i) provide a clear definition of each water mass class, and (ii) ensure that the naming and classification are consistent between Figures 3 and 4. This would greatly help the reader follow the water-mass analysis throughout the manuscript.

Figure 4 : To ensure consistency with the text, it would be helpful to add "W" to the SA and NA labels in the figure to explicitly indicate South Atlantic Water and North Atlantic Water fractions. In addition, the legend should define all water-mass classes so that it fully explains the figure on its own.

l.217 : The Burger number is introduced but its physical meaning or relevance is not explained. Adding a brief sentence describing what this parameter represents (e.g., the relative importance of stratification vs. rotation in setting eddy vertical structure) would help readers understand why it is included.

Figure 5 : As for Figure 2, adding the key isopycnals and the mixed layer depth along the glider sections would greatly help relate the velocity and hydrographic structures to the underlying density field. Including an arrow indicating the glider's direction of travel would also improve the readability of the section.

Figure 6 is difficult to interpret in its current form and would benefit from substantial clarification; please see Major Comment C3 for detailed suggestions.

Figure 7 does not convincingly support the authors' dynamical interpretation; please see Major Comment C1 for a detailed discussion of the limitations of this diagnostic.

l.250 : The good agreement between glider-derived dynamic height anomaly and AVISO ADT (shown in the Supplementary Material) is an important result validating the glider's ability to capture the baroclinic component of mesoscale structures. Given its relevance to the main dynamical argument, this comparison would be more appropriately included in the main text.